# Erythroid Differentiation Regulator 1 Strengthens TCR Signaling by Enhancing PLCγ1 Signal Transduction Pathway

**DOI:** 10.3390/ijms23020844

**Published:** 2022-01-13

**Authors:** Myun Soo Kim, Dongmin Park, Sora Lee, Sunyoung Park, Kyung Eun Kim, Tae Sung Kim, Hyun Jeong Park, Daeho Cho

**Affiliations:** 1Kine Sciences, 525, Seolleung-ro, Gangnam-gu, Seoul 06149, Korea; mskim@kinesciences.com (M.S.K.); dmpark@kinesciences.com (D.P.); srlee@kinesciences.com (S.L.); sypark@kinesciences.com (S.P.); dermacmc@naver.com (H.J.P.); 2Department of Cosmetic Sciences, Sookmyung Women’s University, Cheongpa-ro 47-gil 100 (Cheongpa-dong 2ga), Yongsan-gu, Seoul 04310, Korea; kyungeun@sookmyung.ac.kr; 3Division of Life Sciences, College of Life Sciences and Biotechnology, Korea University, 5-ga, Anam-dong, Seongbuk-gu, Seoul 02841, Korea; tskim@korea.ac.kr; 4Institute of Convergence Science, Korea University, Anam-ro 145, Seongbuk-gu, Seoul 02841, Korea

**Keywords:** Erdr1, TCR signal modulation, PLCγ1

## Abstract

Erythroid differentiation regulator 1 (Erdr1) has previously been reported to control thymocyte selection via TCR signal regulation, but the effect of Erdr1 as a TCR signaling modulator was not studied in peripheral T cells. In this report, it was determined whether Erdr1 affected TCR signaling strength in CD4 T cells. Results revealed that Erdr1 significantly enhanced the anti-TCR antibody-mediated activation and proliferation of T cells while failing to activate T cells in the absence of TCR stimulation. In addition, Erdr1 amplified Ca^2+^ influx and the phosphorylation of PLCγ1 in CD4 T cells with the TCR stimuli. Furthermore, NFAT1 translocation into nuclei in CD4 T cells was also significantly promoted by Erdr1 in the presence of TCR stimulation. Taken together, our results indicate that Erdr1 positively modulates TCR signaling strength via enhancing the PLCγ1/Ca^2+^/NFAT1 signal transduction pathway.

## 1. Introduction

T cells have central roles in adaptive immunity and are involved in various diseases, including infectious diseases, cancer, and inflammatory diseases. Generally, T cells are activated via TCR signaling and initiated by contact with antigen-presenting cells. At this point, the strength of TCR signaling is one of the major factors identified to control the differentiation, function, and survival of T cells. For example, weak TCR signaling leads to the differentiation of regulatory T (Treg) cells, while strong signaling favorably induces effector T cells [1]. Modulation of TCR signaling is also related to functions of Treg, effector, and memory T cells [2,3,4]. In addition, the fate of thymocytes is decided by the strength of TCR signaling during negative and positive selection in the thymus [5,6].

Among many TCR downstream pathways, the Phospholipase Cγ1 (PLCγ1)/calcium/nuclear factor of activated T cells (NFAT) signal transduction pathway is one of the critical factors identified to regulate T cell development, activation, and differentiation [7]. Calcium signal inhibitors prevent the positive selection of thymocytes [8] and calcium-dependent gene expression in T cells [7,9]. In addition, PLCγ1 dependent modulation of integrin avidity is required for proper T cell activation and function [10]. These studies evidence that it is worth identifying novel molecules to control TCR-induced calcium signaling.

Erythroid differentiation regulator 1 (Erdr1) has been reported as an anti-inflammatory cytokine in many disease models, including rosacea, rheumatoid arthritis, and psoriasis [11,12,13]. Recently, it has been reported that the activation of Treg cells is enhanced by Erdr1 in the presence of TCR stimulation [14]. Erdr1 is known to be expressed in various murine tissues, including the spleen, bone marrow, and thymus [15]. We also previously confirmed the expression of Erdr1 in the murine spleen and thymus by real-time PCR. The Erdr1 expression in the spleen and thymus was ~1.7- and ~1.9-fold higher than in bone marrow, respectively, and the protein was detected in both the cortex and medulla of the thymus [16]. In that report, calcium flux and CD69 expression were induced in thymocytes by Erdr1 in the presence of TCR stimulation, and the effects of Erdr1 were inhibited by a Ca^2+^ chelator [16]. Furthermore, in vivo administration of Erdr1 enhanced the positive selection of late-stage CD4/CD8 double-positive thymocytes, suggesting that Erdr1 strengthens TCR signaling in thymocytes and thereby regulating T cell development [16]. Therefore, we hypothesized that Erdr1 also modulates TCR signaling of mature T cells and affects T cell activation status.

In this study, it is shown that Erdr1 is a novel modulator of TCR signaling transduction. Results revealed that Erdr1 increased the expression of an activation marker, CD69, in CD4 and CD8 T cells and proliferation of CD4 T cells in the presence of TCR stimulation. In addition, it was observed that calcium flux, phosphorylation of PLCγ1, and translocation of NFAT1 into nuclei were enhanced in T cells by Erdr1 with TCR stimuli, suggesting that Erdr1 positively regulates TCR signaling via amplifying the PLCγ1/Ca^2+^/NFAT1 pathway.

## 2. Results

### 2.1. Erdr1 Enhances Activation of T Cells in the Presence of TCR Stimulation

To examine the effects of Erdr1 on T cell responses, primary CD4 T and CD8 T cells were isolated from peripheral lymph nodes of mice and cultivated with various concentrations of Erdr1 in the absence or presence of TCR stimulation for 18 h. As 10 to 100 ng/mL of Erdr1 has induced cell death in CD4 T cells, while T cell viability and Treg cell activation have been increased at 1000 ng/mL [14], 10 to 1000 ng/mL of Erdr1 concentrations were selected to observe T cell responses. After culture, the expression of surface CD69, an activation marker, was determined by flow cytometry. Interestingly, Erdr1 (1000 ng/mL) significantly promoted the expression of CD69 in CD4 T cells in the presence of the TCR stimulus while it had no effect in the absence of the anti-CD3 antibody (Figure 1A). We confirmed that cell viability was not changed in this condition Appendix A). In similar experiments using CD8 T cells, Erdr1 also significantly induced CD69 expression only in the presence of TCR stimulation (Figure 1B), suggesting that Erdr1 enhanced the TCR-mediated activation of T cells.

### 2.2. TCR-Mediated Proliferation of CD4 T Cells Is Promoted by Erdr1

Next, Erdr1 activity for T cell proliferation was investigated because cell division is required for the differentiation of helper T cells and proper functions [17]. To confirm T cell proliferation, CD4 T cells from lymph nodes were labeled with CFSE and cultured with or without Erdr1 in the absence or presence of TCR ligation for 72 h. Flow cytometric data described that TCR-induced T cell proliferation was dramatically increased by Erdr1 (Figure 2A), and the percentage of divided T cells of more than three rounds were significantly higher in the Erdr1 treated group (Figure 2B). These results indicate that Erdr1 enhances TCR-dependent proliferation of T cells as well as activation.

### 2.3. Erdr1 Strengthens TCR-Mediated Ca^2+^ Influx and PLCγ1 Phosphorylation in CD4 T Cells

As Erdr1 enhanced the activation and proliferation of T cells (Figure 1 and Figure 2) and Erdr1 induced Ca^2+^ influx in thymocytes [16], it was examined whether Erdr1 changed the cytosolic level of Ca^2+^ in T cells. We used CD4 T cells for signaling studies because CD8 T cells were relatively insensitive to the CD3 antibody-mediated activation (Appendix A). When Fluo-4 loaded CD4 T cells were stimulated with the anti-CD3ε antibody, Erdr1 dramatically enhanced Ca^2+^ influx in CD4 T cells compared to the TCR stimulation alone group (Figure 3A), suggesting that Erdr1 actively tuned the TCR signaling pathway. As the Ca^2+^ influx assay was performed in the calcium-free solution, the increase of cytosolic Ca^2+^ should be due to Ca^2+^ release from the endoplasmic reticulum. As activation of PLCγ1 was required for the release of Ca^2+^ from the endoplasmic reticulum [18], the level of phosphorylated PLCγ1 in CD4 T cells was also determined with the western blot analysis. As expected, the phosphorylation of PLCγ1 was significantly promoted by Erdr1 in the presence of TCR signaling, while it was not altered in the absence of TCR signaling (Figure 3B). These results revealed that Erdr1 amplifies TCR-induced Ca^2+^ influx and PLCγ1 phosphorylation in CD4 T cells.

### 2.4. Erdr1 Increases Translocation of NFAT1 into Nuclei in CD4 T Cells with TCR Stimulation

We then evaluated the translocation of NFAT1 into nuclei because NFAT activation is downstream of calcium signal, and NFAT also mediates CD69 expression in T cells [19,20,21]. CD4 T cells from lymph nodes were activated with TCR stimuli with or without Erdr1 for up to 60 min, and fixed cells were fluorescently stained for NFAT1, nuclei, and actin. NFAT1 translocation was then determined by counting cells using confocal microscopy. Results revealed that NFAT translocation was not induced in the TCR stimulation alone group, probably due to the use of relatively low concentration of anti-CD3 antibody (250 ng/mL), as 5 to 10 mg/mL of the antibody used for TCR signaling and T cell differentiation studies [22,23]. As a low dose of antigen delays NFAT translocation into nuclei [24], the dose of CD3 antibody in this experiment seems too weak to induce translocation in 60 min. In this condition, however, results showed that NFAT1 translocation was significantly increased by Erdr1 in the presence of TCR stimulation (Figure 4), demonstrating that Erdr1 modulates TCR responses via the amplification of the PLCγ1/Ca^2+^/NFAT signaling pathway.

## 3. Discussion

In this study, the TCR-mediated expression of CD69, an activation marker, and proliferation of T cells was enhanced by Erdr1. Results revealed that Ca^2+^ influx in CD4 T cells in response to TCR stimulation was also strengthened in the presence of Erdr1. In addition, phosphorylation of PLCγ1 and NFAT1 translocation into nuclei, which were known to promote the expression of CD69 in T cells [20,21], were significantly induced by Erdr1. Therefore, our results indicate that Erdr1 acts as a TCR signal modulator which amplifies the TCR-mediated PLCγ1/Ca^2+^/NFAT signaling pathway.

It is difficult to examine a molecular mechanism of Erdr1 as a TCR signal modulator because no receptor for Erdr1 has been identified so far, and numerous pathways to strengthen PLCγ exist. PLCγ subtypes are usually activated by receptor tyrosine kinases (RTKs) [25]. However, it is controversial whether RTK signals can enhance TCR signal because RTKs such as insulin receptors and PDGFR induce T cell functions and proliferation [26,27], while VEGFR suppresses T cell activation [28]. The involvement of G protein coupled receptor (GPCR) signals can also be considered for the effects of Erdr1 because the CXCL12 signal through CXCR4 acts as a costimulatory signal for T cells [29]. However, it is not clear whether RTKs and GPCRs mediate the PLCγ1-activating effect of Erdr1 because Erdr1 without TCR stimulation failed to induce the phosphorylation of PLCγ1 (Figure 3). As Erdr1 amplified PLCγ1/Ca^2+^/NFAT signals in the presence of TCR stimuli (Figure 3 and Figure 4), molecules related to the TCR-Ca^2+^ stream, such as CD3, CD4, CD5, and LAT [30], could be mediators for the effect of Erdr1 on TCR signals. There also exist various upstream molecules involved in PLCγ1/Ca^2+^/NFAT signal. Vav mediates the activation of PLCγ1 [31], as well as LAT which is required for initiation of intracellular calcium flux [32]. In addition, phosphorylated ZAP-70 organizes upstream signals of PLCγ1 [31]. Therefore, the effects of Erdr1 on those upstream signal molecules should be further investigated for a better understanding of Erdr1 as a TCR modulator.

Membrane lipids are also candidates as Erdr1 receptors because lipid raft association is required for TCR signals, and regulation of TCR stimulation via controlling lipid rafts aggregation has been reported [33,34,35]. Indeed, Erdr1 was initially found to be localized at the inner side of the plasma membrane in activated T cells, and the T cells showed a phenomenon of Erdr1 disappearance at the site of cell-to-cell contact [15]. In addition, there exists a report describing that Erdr1 expression is increased in brains from TUB gene-mutated mice [36]. It is interesting because Tubby protein, a membrane-bound protein, acts as a transcription factor and enhances T cell activation in a PLCγ dependent manner [37]. From these studies, it is conjectured that components of the cell membrane may be involved in the TCR signal modulating the function of Erdr1. Further studies to discover Erdr1-interacting molecules will provide a better understanding of Erdr1-related T cell immune regulation.

In the previous reports, it has been shown that Erdr1 is highly expressed in T cell-rich organs such as the spleen and thymus and TCR signal strength of thymocytes is regulated by Erdr1 [16]. In the current study, Erdr1 also enhanced Ca^2+^ flux, PLCγ1 phosphorylation, and NFAT1 activation in peripheral CD4 T cells in the presence of TCR stimulation (Figure 3 and Figure 4), suggesting that Erdr1 directly controls T cell immunity via the modulation of TCR signaling. Furthermore, Erdr1 enhanced activation markers for Treg cells and collagen-induced arthritis was ameliorated by Erdr1 via Treg activation [14]. Taken together with the results in the current study, Erdr1, which controls T cell development and the activation of T cells via the modulation of TCR signaling [16], is suggested as a crucial regulator for T cell immunity. Therefore, uses of Erdr1for immune-related diseases via modulating T cell responses could be considered.

## 4. Materials and Methods

### 4.1. Mice and Cells

Eight to 10-week-old female C57/BL6 (OrientBio, Seongnam city, Korea) mice were used. The mice were maintained in a specific pathogen-free facility, and all experiments were approved by Korea University Institutional Animal Care and Use Committee (KUIACUC-2020-0077). T cells were isolated from peripheral lymph nodes and cultured in RPMI 1640 media (WelGENE Inc, Daegu, Korea) with FBS (10 %, WelGENE Inc), penicillin (100 U/mL, Invitrogen, Carlsbad, CA, USA), streptomycin (0.1 mg/mL, Invitrogen), and 2-ME (50 μM, Sigma-Aldrich, St Louis, MO, USA).

### 4.2. Antibodies and Recombinant Erdr1

PerCP-Cy5.5-conjugated anti-CD4 (RM4-5), anti-CD3ε (145-2C11), and anti-hamster IgG1 (G94-56) were purchased from BD Biosciences (San Diego, CA, USA). APC-conjugated anti-CD69 (H1.2F3) was obtained from eBioscience (San Diego, CA, USA). Alexa488-conjugated anti-CD4 (GK1.5) was obtained from BioLend (San Diego, CA, USA). Anti-pPLCγ1 and anti-NFAT1 were obtained from Cell Signaling Technology (Danvers, MA, USA). Anti-PLCγ1 was purchased from Santa Cruz Biotechnology (Dallas, TX, USA).

Recombinant Erdr1 was prepared as described previously [38]. In brief, the bacterial vector with the Erdr1 CDS region was constructed from the Erdr1-pCMV-SPORT6 plasmid (Open Biosystems, Huntsville, AL, USA), and then Erdr1 was purified with over 95% purity from bacteria. Lots with low endotoxin levels (<0.1 EU/mL) determined by the *Limulus Amebocyte* lysate assay (Cape Cod, East Falmouth, MA, USA) were used for experiments.

### 4.3. T Cell Stimulation, Proliferation, and Ca^2+^ Influx Assay

Murine CD4 or CD8 T cells were isolated using T cell isolation kits according to the manufacturer’s instruction (Miltenyi Biotec, Auburn, CA, USA). T cells or CFSE-labled (1 μM) CD4 T cells (1 × 10^5^ cells/well) were incubated with various concentrations of Erdr1 in an anti-CD3ε antibody-coated (125 or 250 ng/mL) 96-well plate. CD69 expression and proliferation were examined by flow cytometry after 18 h and 72 h cultivation, respectively.

For of Ca^2+^ influx assay, Fluo-4 (3 μM, TEF Labs, Inc., Austin, TX, USA) was loaded into CD4 T cells for 40 min at 37 °C, and then cells were rested with anti-CD3ε antibody (2 μg/mL, hamster IgG1) at RT for 15 min. Ca^2+^ influx was detected by flow cytometry immediately after the addition of the anti-hamster IgG1 antibody (5 μg/mL) in the absence or presence of Erdr1 (1 μg/mL).

### 4.4. Flow Cytometry

Single-cell suspensions were stained with antibodies at 1:250 to 1:500 dilution in FACS buffer. Cell surface stain was performed for 15 min at RT. Fixation and intracellular stain were performed using Cytofix/Cytoperm and Perm/Wash solutions (BD Biosciences). Non-specific binding was monitored using fluorescent-conjugated control antibodies. For the Ca^2+^ influx assay, base levels were determined for 30 s without stimuli, and then stimuli were added. Flow cytometric analysis was performed with live cell gates with FACSCalibur and CellQuest or FlowJo software (BD Biosciences).

### 4.5. Western Blot Analysis

Isolated CD4 T cells (2 × 10^6^ cells/group) were incubated for 10 min in the absence or presence of the anti-CD3ε antibody (2 μg/mL, hamster IgG1) at RT. Anti-hamster IgG1 (10 μg/mL) with or without Erdr1 (2.5 μg/mL) was added and cultured for 4 min. The cells were then treated with lysis buffer, and the whole-cell lysates were prepared by 20 min centrifugation at 20,000 g. Proteins were separated by SDS–PAGE and transferred to nitrocellulose membranes. The membranes were blocked with 5% skim milk for 1 h and incubated with primary antibodies (1:5000 dilution) overnight at 4 °C. HRP-conjugated secondary antibodies were treated for 1 h at RT, and protein bands were visualized with HRP substrate (Millipore Corporation, Billerica, MA, USA) using LAS-3000 (Fuji Photo Film Co., Ltd., Tokyo, Japan). The intensity of the bands was evaluated using ImageJ software.

### 4.6. NFAT Translocation

CD4 T cells (2 × 10^6^ cells/group) were cultivated in an anti-CD3ε antibody-coated (250 ng/mL) 24-well plate in the absence or presence of Erdr1 (1 μg/mL) for 30 or 60 min. T cells were fixed in 4% paraformaldehyde and permeabilized with staining buffer (0.5% Triton X-100 solution containing 5% FHS). After blocking with staining buffer for 1 h at RT, T cells were incubated with NFAT1 antibody for 2 h at RT. Cells were then stained with the Alexa488-conjugated secondary antibody, Alexa647-conjugated phalloidin, and DAPI for 1 h at RT. NFAT translocation in CD4 T cells was visualized by confocal microscopy (LSM700, Carl Zeiss, Jena, Germany).

### 4.7. Statistical Analyses

A non-paired and two-tailed Student’s *t*-test was used to compare the control and experimental groups. *p*-values < 0.05 were considered to be statistically significant.

## Figures and Tables

**Figure 1 ijms-23-00844-f001:**
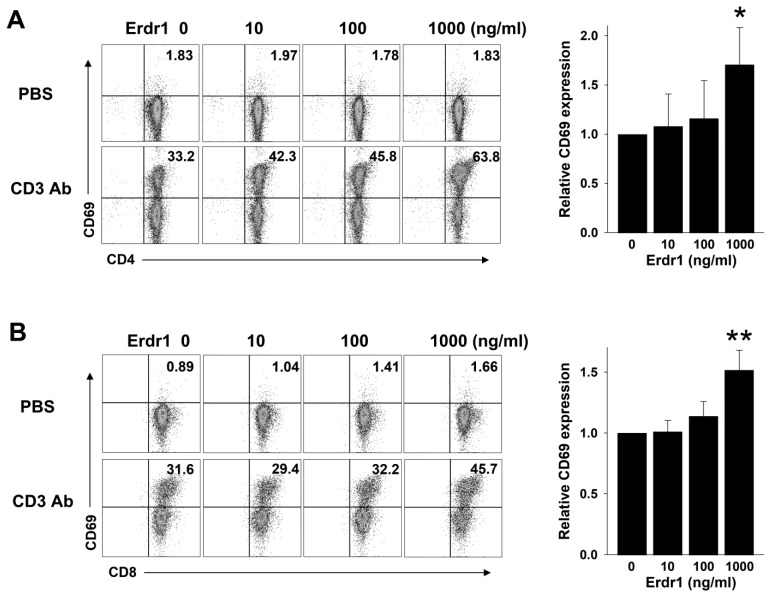
Erdr1 enhances the expression of CD69 in T cells in the presence of TCR stimulation.CD4 T and CD8 T cells were isolated from peripheral lymph nodes of mice and cultivated with the indicated concentrations of Erdr1 in the presence of TCR stimulation for 18 h. (**A**) CD4 T cells were cultured in anti-CD3ε antibody-coated plates (125 ng/mL), and CD69^+^ cells were evaluated by flow cytometry. Results with TCR stimulation were summarized as relative levels of CD69 expression (mean ± SD). (**B**) CD8 T cells were cultured as the same method of the CD4 T cell incubation except 250 ng/mL of anti-CD3 antibody was coated. The expression of CD69 was also examined, and relative levels in the presence of TCR stimulation were summarized as mean ± SD. The numbers indicate the percentages of quadrants. The flow cytometric results are representative of data from three independent experiments. * *p* < 0.05, ** *p* < 0.01 (vs. Erdr1 0 ng/mL).

**Figure 2 ijms-23-00844-f002:**
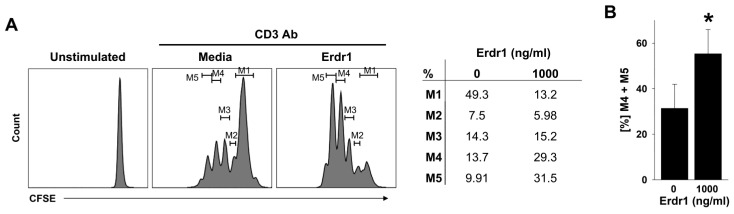
Erdr1 enhances the proliferation of CD4 T cells. (**A**) CD4 T cells from peripheral lymph nodes were labeled with CFSE (1 μM) and cultivated with or without 1 μg/mL of Erdr1 in the presence of the anti-CD3ε antibody (500 ng/mL). The unstimulated control was incubated without the anti-CD3 antibody. After 72 h cultivation, T cell proliferation was evaluated by the flow cytometric analysis. (**B**) Two of the most divided populations (M4 and M5) were summarized as mean ± SD. The flow cytometric results are representative of data from three independent experiments. * *p* < 0.05 (vs. Erdr1 0 ng/mL).

**Figure 3 ijms-23-00844-f003:**
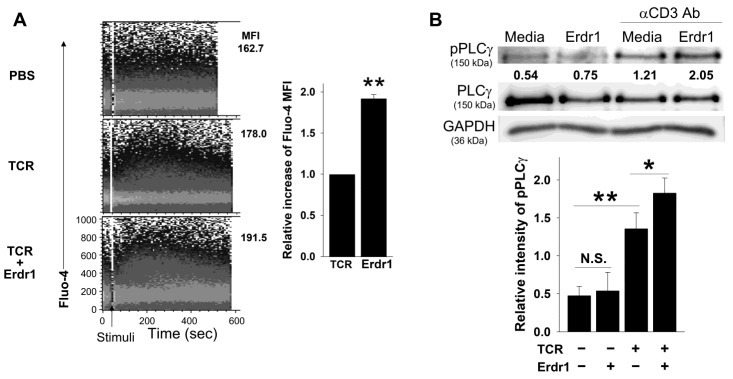
Erdr1 amplifies the TCR-mediated calcium influx and phosphorylation of PLCγ1 in CD4 T cells.CD4 T cells were isolated from peripheral lymph nodes of mice. (**A**) Fluo-4 was loaded into CD4 T cells (3 μM), and cells were incubated with anti-CD3ε antibody (2 μg/mL, hamster IgG1) at RT. Cytosolic Ca^2+^ levels were determined by flow cytometry immediately after the addition of anti-hamster IgG1 antibody (5 μg/mL) in the absence or presence of Erdr1 (1 μg/mL). Relative increase of Fluo-4 MFI was summarized as mean ± SD from three independent experiments. (**B**) CD4 T cells were incubated for 10 min in the absence or presence of anti-CD3ε antibody (2 μg/mL), and the secondary antibody (10 μg/mL) with or without Erdr1 (1 μg/mL) was added. After cultivation for 4 min, the levels of pPLCγ1 in the lysates were measured by western blot analysis. Numbers indicate relative intensity of pPLCγ1 to a loading control, evaluated using ImageJ software, and the data were summarized as mean ± SD. Data of Ca^2+^ influx and western blot bands are representative of three independent experiments. * *p* < 0.05, ** *p* < 0.01, N.S.: not significant.

**Figure 4 ijms-23-00844-f004:**
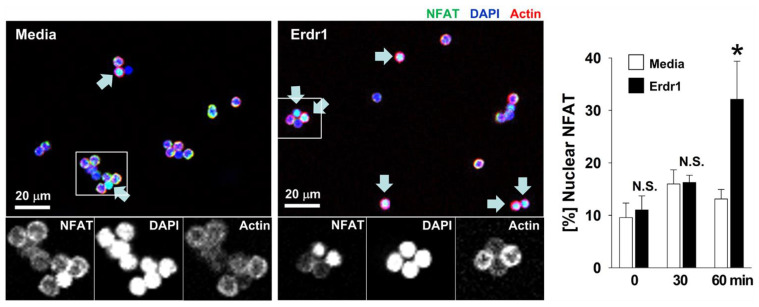
Erdr1 induces translocation of NFAT1 into nuclei in the presence of TCR stimulation.CD4 T cells were isolated from peripheral lymph nodes of mice. The isolated CD4 T cells were cultured in anti-CD3ε antibody (250 ng/mL)-coated plates in the absence or presence of Erdr1 (1 μg/mL) for 30 or 60 min. Cells were then fixed, permeabilized, and stained against NFAT1, followed by phalloidin and DAPI stain. NFAT1 translocation into nuclei was visualized by confocal microscopy, and percentages of NFAT1 translocated cells versus total cells were summarized as mean ± SD. Data are representative of three independent experiments. * *p* < 0.05.

## Data Availability

Not applicable.

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
