# Peer review of "Erythroid Differentiation Regulator 1 Strengthens TCR Signaling by Enhancing PLCγ1 Signal Transduction Pathway"

_ijms, 2022, doi:10.3390/ijms23020844_

Round 1

Reviewer 1 Report

Comments

This study investigated the role of Erdr1 in T cell development and differentiation, which is an extension of the authors’ previous work. The authors found PLCg1/Ca2+/NFAT1 as a potential pathway downstream of Erdr1 in modulating the TCR signaling. The results provide adequate evidence to support a positive role of Erdr1 in TCR activation via promoting the activation of the PLCg1/Ca2+/NFAT1 pathway in T cell activation and proliferation in vitro.

A few comments:

In the introduction:

  1. Line 42~45 reads confusing. Please double check and make corrections if needed.
  2. The level of expression of Erdr1 in lymphoid tissues should be introduced. If it is unclear, this fact should be stated as well.
  3. Line 48~54 is overly too brief. Regarding the known roles of Erdr1 in thymocytes, T cells, and TCR signaling, more details of the known phenotypes and signaling correlates should be described.

For 2.1. Erdr1 enhances activation of T cells in the presence of TCR stimulation (Fig. 1&2)

  1. Erdr1 can function as an autocrine factor to induce apoptosis through caspase 3. Adding Erdr1 to T cells in vitro may cause cell death and so affect the percentages of any phenotypic cell subsets. To better understand whether the percentage of CD69-positive cells is a result of better TCR activation or more cell death of the CD69-negative cells, a serial concentration of Erdr1 should be used for viability testing.
  2. If viability data is not available, the cell numbers of CD4+ and CD8+ T cells after TCR stimulation should help provide information of cell death. The authors should at least check the cell numbers of T cells stimulated by Erdr1.
  3. A minor comment: CD25 is a good activation marker for CD4+ T cells and should be included in the activation experiments if possible.

2.3. Erdr1 strengthens TCR-mediated Ca2+ influx and PLCg1 phosphorylation in CD4 T cells (Fig. 3)

  1. Please indicate the molecular weights of the three targets on the western blotting image. .

2.4. Erdr1 increases translocation of NFAT1 into nuclei of CD4 T cells with TCR stimulation (Fig.4)

  1. Scale bars in the images should be provided.

Author Response

  1. (Comment): Line 42~45 reads confusing. Please double check and make corrections if needed.

(Response): As the reviewer commented, we corrected those sentences (Line 42 to 44).

  1. (Comment): The level of expression of Erdr1 in lymphoid tissues should be introduced. If it is unclear, this fact should be stated as well.

(Response): As the reviewer commented, we added the sentence (Line 51 to 52).

  1. (Comment): Line 48~54 is overly too brief. Regarding the known roles of Erdr1 in thymocytes, T cells, and TCR signaling, more details of the known phenotypes and signaling correlates should be described.

(Response): As the reviewer commented, we added more information about the functions of Erdr1 in thymocytes and TCR signaling (Line 52 to 56).

  1. (Comment): Erdr1 can function as an autocrine factor to induce apoptosis through caspase 3. Adding Erdr1 to T cells in vitro may cause cell death and so affect the percentages of any phenotypic cell subsets. To better understand whether the percentage of CD69-positive cells is a result of better TCR activation or more cell death of the CD69-negative cells, a serial concentration of Erdr1 should be used for viability testing.

(Response): As the reviewer commented, because Erdr1 can induce apoptosis of T cells (Soto, et. al., 2017 PNAS), we also confirmed cell viability during the flow cytometric analysis for CD69 expression. Results showed that Erdr1 did not affect the viability in this condition. We added the results as supplementary materials (Line 74 to 75 and Supplementary Figure S1).

  1. (Comment): If viability data is not available, the cell numbers of CD4+ and CD8+ T cells after TCR stimulation should help provide information of cell death. The authors should at least check the cell numbers of T cells stimulated by Erdr1.

(Response): We added the viability results as supplementary materials (Line 74 to 75 and Supplementary Figure S1).

  1. (Comment): A minor comment: CD25 is a good activation marker for CD4+ T cells and should be included in the activation experiments if possible.

(Response): We agree with the reviewer comment. Evaluation of CD25 expression must be helpful, and we are working on it. Unfortunately, it takes more time for delivery of mice because of Covid-19, so we need more time.

  1. (Comment): Erdr1 strengthens TCR-mediated Ca2+ influx and PLCg1 phosphorylation in CD4 T cells (Fig. 3). Please indicate the molecular weights of the three targets on the western blotting image.

(Response): As the reviewer commented, we indicated the molecular weights of the bands (Figure 3).

  1. (Comment): Erdr1 increases translocation of NFAT1 into nuclei of CD4 T cells with TCR stimulation (Fig.4). Scale bars in the images should be provided.

(Response): As the reviewer commented, we added scale bars in the figure (Figure 4).

Reviewer 2 Report

In this manuscript the authors have investigated the role of Edr1, which thay had previously reported to modulate thymocyte development by enhancing TCR signaling, in the activation of peripheral T cells. They show that Edr1 enhances T cell activation and proliferation by strengthening PLCgamma1 activation and intracellular Ca2+ mobilization, which result in enhanced nuclear translocation of the transcription factor NFAT.

Although this is an extension of previous work by the authors, the results identify a new modulator of TCR signaling, which is interesting considering the impact of the strength of TCR signaling on T cell fate. Overall the data are solid, although the mechanism underlying the TCR-modulating activity of Edr1 has not been characterized (partly due, as the authors state, to the lack of knowledge of the Edr1 receptor).

Specific points

Point 1. In figure 1 the authors tested Edr1 on the TCR-dependent proliferation of both CD4 and CD8 T cells. They then focused on CD4 T cells. Why? TCR signal strength may influence differently signaling in CD8 versus CD4 T cells.

Point 2. Related to point 1, why have the authors used different concentrations of anti-CD3 mAb to activate CD4 and CD8 T cells (125 ng/ml vs 250 ng/ml)?

Point 3. The biologically active concentration of Edr1 in the various assays is 1000 ng/ml. This appears very high. Can the authors comment on this point?

Point 4. Figure 3. The Ca2+ flux data should be quantified over multiple experiments and plotted (with stats). Additionally, it is not clear why the authors have focused solely on the PLCgamma1/Ca2+/NFAT pathway. Could Edr1 also affect TCR signaling upstream? To better delineate the mechanism of TCR modulation by Edr1 it would be useful to test whether CD3zeta phosphorylation and LAT phosphorylation are affected.

Point 5. Figure 4. The magnification of the image is too low. The cells indicated with an arrow could be shown at higher magnification. Regarding the histogram, it appears that stimulation of the TCR alone does not induce NFAT translocation at neither of the two activation times. This is strange, since TCR cross-linking is sufficient to elicit Ca2+ flux and NFAT nuclear translocation.

Author Response

  1. (Comment): In figure 1 the authors tested Edr1 on the TCR-dependent proliferation of both CD4 and CD8 T cells. They then focused on CD4 T cells. Why? TCR signal strength may influence differently signaling in CD8 versus CD4 T cells.

(Response): As the reviewer commented, it is meaningful to study whether Erdr1 affect TCR signaling in CD8 T cells as the same way in CD4 T cells. The effects of Erdr1 on CD8 T cells must be further studied and we are working on it. Indeed, CD8 T cells are relatively insensitive to CD3e antibody-mediated activation (Crespo, et. al., 2021 PLoS One), and our results also showed a similar phenomenon (Supplementary Figure S2). These made it difficult for us to select experimental conditions for signaling studies in CD8 T cells. With these reasons, we first tried to determine the effects of Erdr1 on TCR signaling in CD4 T cells. We added comments about this in the Results section and supplementary materials (Line 109 to 111, Supplementary Figure S2).

  1. (Comment): Related to point 1, why have the authors used different concentrations of anti-CD3 mAb to activate CD4 and CD8 T cells (125 ng/ml vs 250 ng/ml)?

(Response): As answered in the “Comment 1”, CD8 T cells are relatively insensitive to CD3e antibody-mediated activation (Crespo, et. al., 2021 PLoS One), so used 250 ng/ml of anti-CD3 Ab for CD8 T cells.

  1. (Comment): The biologically active concentration of Edr1 in the various assays is 1000 ng/ml. This appears very high. Can the authors comment on this point?

(Response): We agree with the reviewer comment. There is no defined biological activity unit for Erdr1 so far. However, Erdr1 seems to act on T cell activation at high concentrations, since the thymocyte activation was affected by 0.5 to 5 mg/ml of Erdr1 in our previous report (Kim, et. al., 2019 Cell Immunol). As Erdr1 is a multifunctional protein which affects various cell types as an apoptotic factor, an anti-inflammatory cytokine, and a T cell regulator (Houh, et. al., 2016 Int J Mol Sci; Park, et. al., 2020 Int J Mol Sci; Kim, et. al., 2019 Cell Immunol; Kim, et. al., 2020 Int J Mol Sci), this cytokine may work quite differently according to concentrations.

  1. (Comment): Figure 3. The Ca2+ flux data should be quantified over multiple experiments and plotted (with stats). Additionally, it is not clear why the authors have focused solely on the PLCgamma1/Ca2+/NFAT pathway. Could Edr1 also affect TCR signaling upstream? To better delineate the mechanism of TCR modulation by Edr1 it would be useful to test whether CD3zeta phosphorylation and LAT phosphorylation are affected.

(Response): As the reviewer commented, we changed the data as density plots and total MFI values of Fluo-4 were indicated (Fig. 3A). Relative increase of MFI from three independent experiments was also summarized (Fig. 3A). We agree with the additional comment. The other TCR-mediated signaling molecules should be further investigated with Erdr1 and we are working on it. As numerous factors are involved in TCR signal, we started this study with focus on calcium signal, which was previously found to be affected by Erdr1 in thymocytes (Kim, et. al., 2019 Cell Immunol).

  1. (Comment): Figure 4. The magnification of the image is too low. The cells indicated with an arrow could be shown at higher magnification. Regarding the histogram, it appears that stimulation of the TCR alone does not induce NFAT translocation at neither of the two activation times. This is strange, since TCR cross-linking is sufficient to elicit Ca2+ flux and NFAT nuclear translocation.

(Response): As the reviewer commented, we added images with higher magnification and separated fluorescent signals (Fig. 4). About the TCR alone group, the experiments were performed with a relatively low concentration of anti-CD3 Ab (250 ng/ml) to show Erdr1 effects, as 5 - 10 mg/ml of CD3 Ab was used for TCR signaling studies and T cell differentiation (Hartl, et. al., 2020 Nat. Immunol.; He, et. al., 2018 PNAS). Also, T cell activation protocols from antibody vendors suggest using 5 – 10 mg/ml of CD3 Ab (Thermofisher, https://tools.thermofisher.com/content/sfs/manuals/t-cell-activation-in-vitro.pdf; BioLegend, https://www.biolegend.com/en-us/protocols/t-cell-activation-with-anti-cd3-antibodies-protocol-mouse). As T cell stimulation with a low dose of antigen shows delayed NFAT translocation into nuclei (Lodygin, et. al., 2013 Nat Med), the low concentration of anti-CD3 Ab in our experiments seems weak to induce NFAT translocation in 60 min.

Round 2

Reviewer 1 Report

In the introduction:

Line 50~51 is still brief. Please supply more detailed information.

For 2.1. Erdr1 enhances activation of T cells in the presence of TCR stimulation (Fig. 1&2):

A minor comment: Cell viability and activation results with higher Erdr1 concentration ( > 1000ng/ ml) should be provided.

Author Response

  1. (Comment): Line 5~51 is still brief. Please supply more detailed information.

(Response): As the reviewer commented, we added more information in the “Introduction” section (Line 51 to 56).

  1. (Comment): For 2.1. Erdr1 enhances activation of T cells in the presence of TCR stimulation (Fig. 1&2):

A minor comment: Cell viability and activation results with higher Erdr1 concentration ( > 1000ng/ ml) should be provided.

(Response): As the reviewer commented, results with higher concentrations of Erdr1 should provide us much information. Unfortunately, it takes more time to perform additional experiments. In our previous report (Kim, et. al., 2020, Int J Mol Sci), similar experiments have been performed with prolonged culture time (48h). Results showed that cell viability was decreased at low concentrations (10 and 100 ng/ml) of Erdr1 in the absence of TCR stimuli, while it was increased at 1000 ng/ml both in the absence and presence of TCR stimulation (Figure in attached file). The results suggest that Erdr1 may act as apoptotic factor at relatively lower concentrations and Erdr1 does not affect T cell viability in short time cultivation from the current study (Supplementary Figure S1). We added the reason we selected the doses of Erdr1 in the “Results” section (Line 74 to 77).

Reviewer 2 Report

Overall the authors have addressed satisfactorily the concerns raised in my previous review. I suggest however that they incorporate the response to point 5 (lack of NFAT translocation to the nucleus in response to TCR stimulation alone) in the Discussion. They should also address at least in the Discussion the point about TCR signals upstream of Ca2+/PLCgamma/NFAT that could be modulated by Edr1 and hence impact on this signaling axis.

Author Response

  1. (Comment): I suggest however that they incorporate the response to point 5 (lack of NFAT translocation to the nucleus in response to TCR stimulation alone) in the Discussion. They should also address at least in the Discussion the point about TCR signals upstream of Ca2+/PLCgamma/NFAT that could be modulated by Edr1 and hence impact on this signaling axis.

(Response): As the reviewer commented, we indicated the information about the TCR alone group in the “Results” section (Line 148 to 153). We also inserted sentences about TCR signaling of upstream Ca2+/PLCgamma/NFAT in the “Discussion” section (Line 185 to 190).